# Non-Destructive Textural Characterization of Southern Romanian Neolithic and Chalcolithic Pottery Using Digital Image Analysis on Tomographically Reconstructed Sections

Daniel Stoicescu [1,2,3,*], Octavian G. Duliu [2,4,5], Vasile Opriş [1,6,*], Bogdan Manea [1], Izabela Mariş [7], Valentina Voinea [8], Pavel Mirea [9], Valentin Parnic [10], Mădălina Dimache [11] and Cătălin Lazăr [1]

1  ArchaeoSciences Division–ICUB, University of Bucharest, 90 Panduri Street, Sector 5, 050663 Bucharest, Romania
2  Doctoral School on Physics, Faculty of Physics, University of Bucharest, 077125 Măgurele Romania; o.duliu@fizica.unibuc.ro
3  Horia Hulubei National Institute for R&D in Physics and Nuclear Engineering, 30 Reactorului Str., 077125 Măgurele, Romania; bogdan.manea@icub.unibuc.ro (B.M.); catalin.lazar@icub.unibuc.ro (C.L.)
4  Department of Structure of Matter, Earth and Atmospheric Physics, Astrophysics, Faculty of Physics, University of Bucharest, 405 Atomiştilor Str., 077125 Măgurele, Romania
5  Geological Institute of Romania, 1, Caransebeş Str., 012271 Bucharest, Romania
6  History Department, Bucharest Municipality Museum, 2, I.C.Bratianu Blvd., 030174 Bucharest, Romania
7  OMV Petrom, Institute of Research and Technological Design, 29, Culturii Blvd., 105600 Câmpina, Romania; izabela.maris@petrom.com
8  National History and Archaeology Museum Constanţa, 12 Piaţa Ovidiu, 900745 Constanţa, Romania; valentina.voinea@minac.ro
9  Teleorman County Museum, 1 1848 Str., No. 1, 140033 Alexandria, Romania; pavelcmirea@yahoo.com
10  Lower Danube Museum Călăraşi, 4 Progresului Str., 910001 Călăraşi, Romania; vgumelnita@yahoo.com
11  National History Museum of Romania, 12 Calea Victoriei, 030026 Bucharest, Romania; madalina.dimache@mnir.ro
*  Correspondence: daniel.stoicescu@nipne.ro (D.S.); vasile.opris@muzeulbucurestiului.ro (V.O.)

**Abstract:** Pottery is a complex archaeological material that is found ubiquitously in various spatial—temporal frameworks from all over the world; therefore, it is of great importance to archaeological research. The current paper aims to present and discuss the results obtained on a batch of Neolithic (ca. 6000–5000 BC) and Chalcolithic (ca. 5000–3900 BC) pottery sherds from Southern Romania through X-ray Computed Tomography, a non-destructive methodology that allows for the 3D reconstruction and precise measurement of inclusions and voids present within ceramic artefacts. Images from several potsherds were subsequently exported and analyzed by means of dedicated software (ImageJ 1.54p and GIMP) to extract quantitative information on the observed features. Grain size and morphometric analyses were performed on the particles, while the contour variability of the examined inclusions was characterized through the application of shape descriptors. Voids were analyzed in order to reveal specific orientation patterns through the examination of the aspect ratio of the holes and of the Rayleigh z test values. These analyses evidenced the general reliance of moderately and poorly sorted clays for ceramic production, accompanied by a gradual transition from organic to grog tempering, while conservative traditions remained prevalent in primary pottery-forming processes.

**Keywords:** Balkan; Neolithic; pottery; archaeometry; X-rays; image analysis

## 1. Introduction

Pottery is of particular importance to archaeologists, as studies based on the typological, technological, and provenance analyses of this type of material enable a better understanding of the relationship between certain past technological approaches and the contemporaneous social and economic trends. These datasets can then be associated with specific communities from certain, well-defined chronological and spatial frameworks, and

contribute to the elaboration of a broader perspective on how traditions evolved, either independently and/or through external influence [1].

Even if its production is based on a long-standing tradition, pottery still holds many valuable secrets that archaeometry can access today. The complex chaîne opératoire involved in pottery manufacturing requires a systematic investigation of the external and internal structure of the sherds to identify some clear characteristics for every operational sequence (harvesting of raw materials, paste preparations, primary and secondary forming techniques). Moreover, during the firing process, the pottery undergoes many physical and chemical transformations [2], becoming a hard, durable material filled with various mineral structures, voids, tempers, and so on.

During the last century, many analytical techniques have been applied to the study of pottery. However, the destructive nature of many of these available methods presents an important issue when handling cultural heritage objects. Fortunately, research technologies have become increasingly sophisticated and accurate, and now offer the possibility of performing non-destructive investigations, which represents an important step in ensuring the preservation of the studied materials. Among these innovative methods, X-ray Computed Tomography (XCT hereafter) creates good quality three-dimensional images that can provide an overview of the internal structure of the analyzed objects, thus revealing multiple features that would be otherwise unidentifiable. Based on the principles of the interactions between X-rays and the examined material, together with an algorithm to reconstruct the acquired data [3–6], this method has been successfully applied in pottery research. Although the first studies of this kind were performed back in the early 1930s [7], the full potential of this method was recognized much later, when it was applied to the identification and analysis of clay, inclusions, voids, cracks, and surfaces [8–16]. These results proved that consistent sets of data could be acquired by analyzing the internal structure of ceramics in a non-destructive manner.

Despite its numerous advantages, conventional XCT suffers from several limitations. The presence of artifacts which appear during the tomographic scanning, such as beam hardening, ring artifacts, and partial volume effects, presents an important issue. A detailed description of these phenomena may be found in several research papers [17,18].

Another significant drawback is that mineral phases with different chemical compositions are difficult to distinguish at a specific energy, due to their similarity between attenuation coefficients. Even if several methodologies exist to mitigate these effects, they have been mainly applied for the analysis of geological samples, which usually present a more complex mineralogy. An alternative way is to combine the results of the mineralogical and chemical analyses collected through elemental mapping and the information obtained from several geometrical descriptors of the particles [19].

Another limitation is directly related to the size of the reconstructed voxel size. Since pottery is a complex material, particles with many sizes and shapes can be found. Hence, the dimensional determination of particles whose size is comparable to the voxel's is limited by the resolution of the imaging system and the associated artifacts. Therefore, several procedures have to be followed to obtain the best results.

### 1.1. Southern Romania Neolithic and Chalcolithic Pottery

The oldest burnt clay vessels in the Southern Romania area date back to about 8000 years ago [20,21] and were created by the first Neolithic communities coming from the south, who followed the Anatolia–Balkans route [22]. The good quality and large quantity of ceramics discovered in the earliest Neolithic sites north of the Danube indicate a local production, made by experienced potters, who were most likely integrating into newly arrived human groups [21]. During the next two millennia, new ceramic types were spread over wide areas, resulting in the formation of specific styles. Over time, the perpetuation and synthesis of this repertoire of shapes and techniques led to the formation of ceramic traditions, the identification of which has played a fundamental role in defining the archaeological "cultures" as we know them today. Within the area of interest,

ceramic traditions designated as the Starčevo-Criş, Dudeşti, Vădastra, Boian, Hamangia, and Gumelniţa "cultures" followed one another during the Neolithic (ca. 6000–5000 BC) and Chalcolithic (ca. 5000–3900 BC) periods. Generally, the literature emphasizes the rich cultural material of these so-called cultures, in particular through the typological analysis (shapes and decoration) of the pottery discovered in archaeological sites, associated with various prehistoric communities.

Over time, the variations observed in the shapes and decorations of ceramic vessels were not always accompanied by technological innovations or radical changes, as the main paste recipes and manufacturing methods were often perpetuated from one "culture" to another. For example, during the Neolithic period, potters mostly used organic temper [23,24], while during the Chalcolithic they seem to have favored mineral temper [25,26].

Regarding the manufacturing methods, there are no clear preferences associated with certain periods. However, there have been a few attempts to tackle this issue tangentially through isolated case studies [27,28], which identified slabs, coiling, pinching, moulding, and composite methods as frequently used primary forming techniques.

During the last decade, several Neolithic and Chalcolithic pottery and clay objects from Southern Romania have been analyzed by XCT and published as case studies [28–31], advancing this field of research by generating new questions and the current investigation.

### 1.2. Objectives

Our paper aims to present the results of an imaging study on pottery fragments from sites in Southern Romania (Figure 1) in order to trace traditions and changes in clay preparation, paste recipes, and manufacturing methods throughout ca. two millennia (Neolithic and Chalcolithic periods).

Twenty-two samples were selected for the present study and subjected to a non-destructive imaging investigation, aiming to examine the inclusions and voids from the internal structure of pottery.

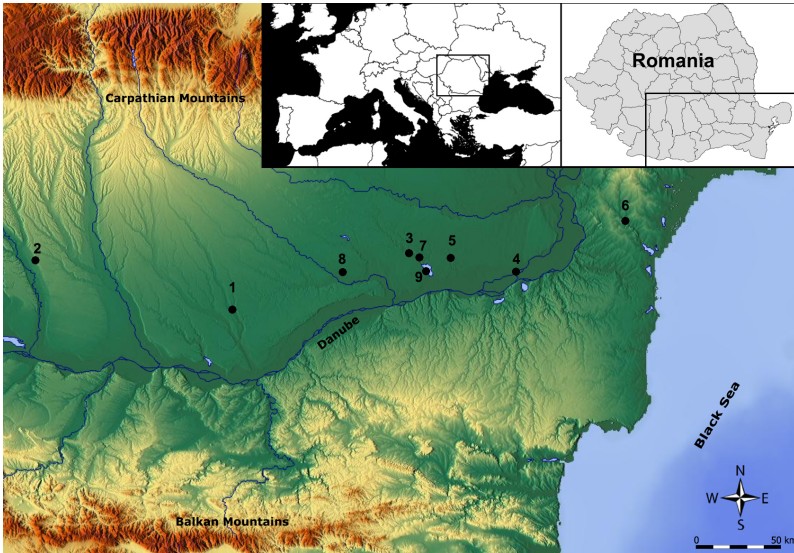

**Figure 1.** The schematic map of the Lower Danube area bearing indications of the Neolithic and Chalcolithic settlements from where pottery fragments were sampled: 1—Măgura Buduiasa; 2—Cârcea Viaduct; 3—Lunca; 4—Grădiștea Coslogeni; 5—Gălățui-Movila Berzei; 6—Cheia; 7—Vlădiceasca; 8—Vidra; 9—Sultana-Malu Roșu.

## 2. Materials and Methods

### 2.1. Pottery Samples

To cover the whole spectrum of ceramic traditions from Southern Romanian Neolithic and Chalcolithic periods, 22 pottery samples from nine archaeological sites were selected for this study, belonging to six archaeological cultures (Table 1). The common feature of all

the samples is the presence of at least one decorative style characteristic of its corresponding archaeological culture. Thus, red sherds painted white or black were selected for the Early Neolithic period, sherds with incised/excised decoration filled with white pigment were chosen for the Middle Neolithic and Early Chalcolithic periods, while for the Developed Chalcolithic, sherds with specific graphite painted motifs were analyzed.

**Table 1.** Pottery samples used for this study which have been scanned through X-ray Computed Tomography.

| Nr. Crt | Sample Name | Construction Zone | Archaeological Site | Period | Culture/Phase | Dating |
|---|---|---|---|---|---|---|
| 1 | P0006 | body | Măgura Buduiasa | Early Neolithic | Starčevo-Criş/I | ca. 6000–5900 |
| 2 | P0023 | body | Măgura Buduiasa | Early Neolithic | Starčevo-Criş/I | ca. 6000–5900 |
| 3 | P0040 | body | Măgura Buduiasa | Early Neolithic | Starčevo-Criş/III | ca. 5800–5500 |
| 4 | P0046 | body | Măgura Buduiasa | Early Neolithic | Starčevo-Criş/III | ca. 5800–5500 |
| 5 | P0073 | body | Măgura Buduiasa | Middle Neolithic | Dudeşti | ca. 5500–5200 |
| 6 | P0082 | body | Măgura Buduiasa | Middle Neolithic | Vădastra | ca. 5200–4800 |
| 7 | P0083 | body | Măgura Buduiasa | Middle Neolithic | Vădastra | ca. 5200–4800 |
| 8 | P0102 | rim | Cârcea Viaduct | Middle Neolithic | Vădastra | ca. 5200–4800 |
| 9 | P0147 | shoulder | Lunca | Middle Neolithic | Boian/I | ca. 5200–4800 |
| 10 | P0149e | body | Lunca | Middle Neolithic | Boian/I | ca. 5200–4800 |
| 11 | P0150a | body | Grădiştea Coslogeni | Middle Neolithic | Hamangia/II | ca. 5200–4800 |
| 12 | P0159 | rim | Grădiştea Coslogeni | Middle Neolithic | Hamangia/II | ca. 5200–4800 |
| 13 | P0173a | body | Gălăţui-Movila Berzei | Middle Neolithic | Boian/II | ca. 5100–4600 |
| 14 | P0177c | body | Gălăţui-Movila Berzei | Middle Neolithic | Boian/II | ca. 5100–4600 |
| 15 | P0112 | rim | Cheia | Early Chalcolithic | Hamangia/III | ca. 5000–4700 |
| 16 | P0116 | body | Cheia | Early Chalcolithic | Hamangia/III | ca. 5000–4700 |
| 17 | P0118d | rim | Vlădiceasca | Early Chalcolithic | Boian/III | ca. 4900–4600 |
| 18 | P0126d | body | Vlădiceasca | Early Chalcolithic | Boian/III | ca. 4900–4600 |
| 19 | P0131 | body | Vidra | Early Chalcolithic | Boian/III | ca. 4900–4600 |
| 20 | P0124a | shoulder | Vlădiceasca | Developed Chalcolithic | Gumelniţa/I | ca. 4600–4300 |
| 21 | P0125b | shoulder | Vlădiceasca | Developed Chalcolithic | Gumelniţa/I | ca. 4600–4300 |
| 22 | P0106 | rim | Sultana Malu Roşu | | Gumelniţa/II-III | ca. 4350–3900 |

### 2.2. X-ray Computed Tomography

All XCT scans of pottery sherds were performed at the Horia Hulubei-National Institute for Physics and Nuclear Engineering (IFIN-HH), using an XTH 225-ST installation from Nikon Metrology. This device is equipped with a microfocus source, which has the following parameters: 225 kV maximum voltage, 1 mA maximum current, and 3 μm focal spot size under 7 W, reaching a maximum value of 225 μm at 225 kV. A flat-panel detector from Varian, with an active area of 1536 × 1920 active pixels, was used for the detection of the X-rays. The pixel pitch size is 127 μm.

The scans on potsherds were performed using the following parameters: 100 kV and 300 μA beam, 354 ms exposure time. A 0.6-mm Al + Cu filter was used to attenuate the soft X-rays. Each object was scanned in 360 steps, representing a full rotation. Table 2 shows the calculated voxel size and source-to-object distance for each scanned sample. CT Pro3D software was used to reconstruct the acquired data. After the acquisition and reconstruction of the images, VGStudio Max 3.0 (Volume Graphics, Heidelberg, Germany) was used to visualize both the 3D and the sections of the objects. The resulting 2D images from the CT scans were exported as TIFF files.

Various complex structures exist in the composition of the fabric of the sherds, making them suitable to be well characterized using X-rays. The images projected are represented by an array of attenuation coefficients, which are correlated in the pictures as grayscale values. Higher values correspond to regions of high-density material, while the areas with low grayscale values correspond to the parts that contain voids or low-density material (Figure 2).

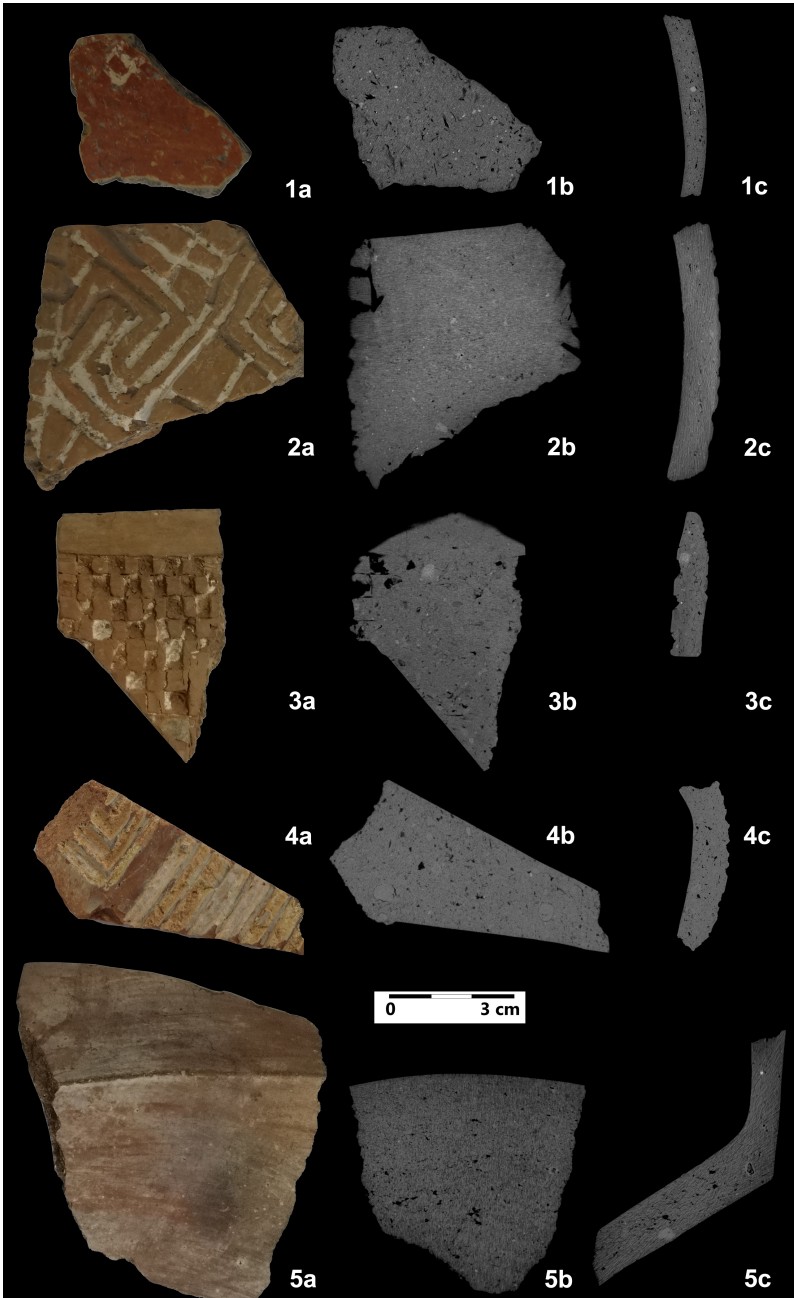

**Figure 2.** Selection of analyzed pottery fragments from Early Neolithic ((**1a**–**1c**); P0023), Middle Neolithic ((**2a**–**2c**); P0082 and (**3a**–**3c**); P0173a), Early Chalcolithic ((**4a**–**4c**); P0126d) and Developed Chalcolithic ((**5a**–**5c**); P0124a). (**a**) Photographic image of the fragment; (**b**) frontal cross section of the tomographic reconstruction of the sample, and (**c**) longitudinal cross section of the tomographic reconstruction of the sample.

### 2.3. Image Analysis

During the visualization with VGStudio Max software, a careful inspection of the reconstructed data was performed. The volume rendering of the samples was also achieved with VGStudio MAX software to obtain a better separation between the clay matrix and the inclusions. Image processing and analysis were carried out by using the GIMP [32] and Fiji programs [33] on several representative samples, based on the large number of inclusions found in the segmented images. A segmentation analysis was performed to enhance the separation between the clay and the several types of inclusions. Characterization of the segmented particles was performed with the Krumbein $\phi$ function of the MinFeret

parameter, using several statistical parameters (determined using the method of moments and the graphic method) and descriptors. In the case of the graphic method, the formulas used to calculate the statistical parameters (standard deviation, skewness, and kurtosis) are the ones developed by Folk and Ward [34]. In contrast, the MinFeret diameter provided by ImageJ analysis was used for phi scale, as suggested by Dal Sasso et al. [19].

**Table 2.** Voxel size and source-to-object distances for pottery samples.

| Nr. Crt | Sample Name | Voxel Size (μm) | Source-to-Object Distance (mm) |
|---|---|---|---|
| 1 | P0006 | 25.71 | 196.97 |
| 2 | P0023 | 35.77 | 274.09 |
| 3 | P0040 | 27.66 | 211.95 |
| 4 | P0046 | 38.43 | 294.45 |
| 5 | P0073 | 38.95 | 298.46 |
| 6 | P0082 | 60.20 | 461.22 |
| 7 | P0083 | 27.78 | 212.23 |
| 8 | P0102 | 63.95 | 489.59 |
| 9 | P0147 | 51.02 | 390.92 |
| 10 | P0149e | 29.30 | 224.44 |
| 11 | P0150a | 40.93 | 313.62 |
| 12 | P0159 | 41.20 | 315.61 |
| 13 | P0173a | 42.98 | 329.34 |
| 14 | P0177c | 79.86 | 611.42 |
| 15 | P0112 | 51.42 | 393.94 |
| 16 | P0116 | 50.49 | 386.85 |
| 17 | P0118d | 30.71 | 235.32 |
| 18 | P0126d | 44.38 | 339.99 |
| 19 | P0131 | 67.08 | 513.95 |
| 20 | P0124a | 60.30 | 462.02 |
| 21 | P0125b | 67.23 | 515.13 |
| 22 | P0106 | 68.87 | 527.14 |

The analysis was conducted following two main axes of investigation. Firstly, the particles characterized by higher grayscale values were segmented from the reconstructed slices using GIMP software (v. 2.10.30, Kimball S. and Mattis P., Măgurele, Romania). A thresholding algorithm was then applied to separate the particles from the background, while the 'Analyze Particles' module was used to perform shape and size analyses on the observed inclusions in six selected samples. Only inclusions with a MinFeret diameter at least 3 times greater than the voxel size were selected to ensure the final data were not affected by XCT and segmentation-generated artifacts.

Since the lowest density areas correspond to the voids, seven representative samples were selected for image processing in order to determine the orientations of the voids. A gamma filter was applied in GIMP to equalize the gray levels between the matrix and inclusions, followed by thresholding and convolution filtering to separate the voids from the rest of the matrix as well as to ease their identification as particles. Gnumeric spreadsheets were used to visualize and examine the acquired data. After the extraction of information in the form of angles, rose diagrams were created to emphasize the most likely orientations of the voids. Rayleigh z test statistic values were calculated for each sample to observe if the voids were uniformly distributed or if they showed preferred orientations.

### 3. Results

*3.1. Inclusion Analysis*

Visualizing of virtual sections offered some insights into the wide range of sizes and shapes of the particles present within the internal structure of pottery fragments. It also provided some partial information on the nature of these particles and on their spatial distribution within the fabric. Based on their grayscale value, most non-plastics could be interpreted as particles with a higher density than the clay matrix. Some trends correspond-

ing to the analytical categories of the archaeological culture framework could be observed: Most of the inclusions found in material of the Starčevo-Criş (I, III), Dudeşti, and Boian (I) pottery sherds have rounded shapes, with various sizes (up to 5 mm) and sorting degrees, which is indicative of natural mineral particles specific to poorly sorted clay sources, and is usually associated with organic tempering. In the sherds from Vădastra, Hamangia (II-III), Boian (II-III), and Gumelniţa (I-III) cultures, besides the moderately or poorly sorted natural inclusions, CT scans and virtual 2D sections emphasize the appearance of grog temper in most of the samples, which has a gray value similar to, or slightly lower than, one of the clays matrices and angular shapes.

Besides the general observations made on the inclusions from all the analyzed samples, after close inspection of the XCT sections, six of them were selected (Early and Middle Neolithic) for some additional dimensional and sorting degree analyses by outlining of the particles using Fiji (see Figure 3).

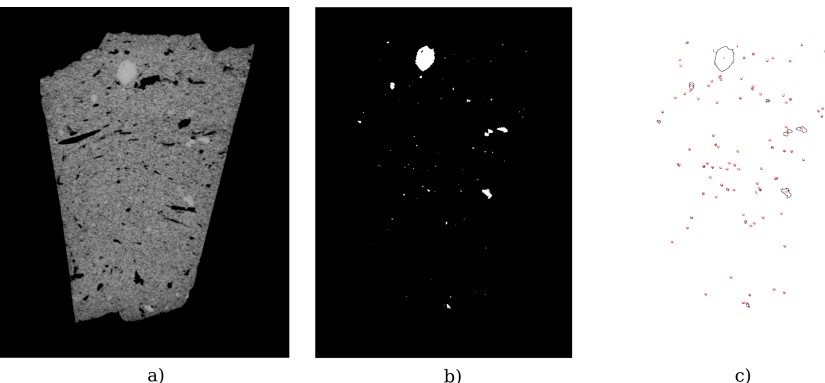

a)                                b)                                c)

**Figure 3.** (**a**) Frontal tomographic section of the P0149e pottery fragment; (**b**) segmented image emphasizing the inclusions; and (**c**) outlines of particles analyzed using Fiji.

As seen in Table 3, the mean values obtained through both methods do not present significant differences when they are framed between 2Φ and 3Φ size classes (or between 250 and 125 μm), which is well emphasized by the bar plots (see Figures S1 and S2). According to the current grain size charts [35], the inclusions analyzed are labeled as fine and very fine sand.

Another relevant parameter is represented by the standard deviation of the grain's population or sorting. Standard deviation values obtained through the method of moments do not differ significantly from the ones obtained through the graphic method. Nevertheless, the description uses the values obtained by the latter method. All samples present values which are framed between 0.30Φ and 0.98Φ (roughly between 800 and 500 μm) on frontal tomographic sections, while the longitudinal sections have values between 0.46Φ and 1.11Φ (roughly between 700 and 500 μm), suggesting that the inclusions are moderately or poorly sorted, according to Jipa and Anastasiu [36].

Skewness values determined for the current samples show significant differences, where those determined by both the method of moments and the graphic method are negative for the frontal sections, thus suggesting that the distributions may point to positive $\phi$ values. The values provided by the graphic method are in most cases positive, except for P0006 (longitudinal section) and P0173a. The distributions of all the particles from the side-view images are negatively skewed, thus pointing mostly to positive phi values (with more fine particles).

**Table 3.** Statistical parameters obtained from the inclusions segmented on images from frontal and longitudinal tomographic sections of the selected samples.

| Sample | Section | Method of Moments | | | | Graphic Method | | | |
|--------|---------|------|----------|------------|-------|------|----------|----------|----------|
| | | Mean | $\sigma$ | Skewness | K | Mean | $\sigma$ | Skewness | Kurtosis |
| P0006 | frontal | 2.69 | 0.37 | −1.22 | 3.54 | 2.65 | 0.30 | −0.56 | 0.86 |
| | longitudinal | 2.97 | 0.55 | −1.00 | 1.68 | 3.02 | 0.46 | −0.03 | 0.91 |
| P0023 | frontal | 2.30 | 0.58 | −0.95 | 0.08 | 2.31 | 0.56 | −0.38 | 0.99 |
| | longitudinal | 2.39 | 0.86 | −2.38 | 7.38 | 2.52 | 0.52 | −0.67 | 0.83 |
| P0046 | frontal | 2.42 | 0.52 | −0.64 | −0.37 | 2.46 | 0.51 | 0.05 | 1.32 |
| | longitudinal | 2.37 | 0.64 | −1.06 | 0.26 | 2.44 | 0.58 | −0.43 | 0.78 |
| P0073 | frontal | 2.40 | 0.67 | −1.80 | 3.90 | 2.46 | 0.56 | −0.47 | 1.10 |
| | longitudinal | 2.44 | 0.57 | −1.61 | 3.15 | 2.48 | 0.52 | −0.43 | 0.93 |
| P0149e | frontal | 2.20 | 1.09 | −1.29 | 1.52 | 2.22 | 0.98 | −0.47 | 0.83 |
| | longitudinal | 2.43 | 1.39 | −1.55 | 1.89 | 2.57 | 1.11 | −0.60 | 1.27 |
| P0173a | frontal | 2.14 | 0.65 | −1.27 | 1.31 | 2.13 | 0.61 | −0.44 | 0.95 |
| | longitudinal | 2.41 | 0.53 | −2.14 | 5.59 | 2.49 | 0.44 | −0.79 | 1.36 |

Kurtosis values are varied from one distribution to another. Also, there are notable differences between the methods employed, as the values obtained through the method of moments have a greater variability within all samples than the ones obtained through the graphic method. The values obtained through the method of moments emphasize the existence of extremely leptokurtic distributions in the case of P0023, P0073, and P0173a, while the rest are framed as platykurtic and mesokurtic. Graphic method values show that all samples present platykurtic to mesokurtic distributions of particles.

Moreover, several trends are observed. In the analyzed samples, a decrease in the mean Phi(MinFeret) is generally observed, while the sorting degree presents an increasing trend, especially in the case of the frontal tomographic sections. This may suggest the existence of changes in the type of clay source used, with a slight tendency towards the use of clays with coarser and a more poorly sorted distribution of inclusions.

Table 4 shows the mean and median values for the shape descriptors: circularity, aspect ratio, and roundness. In almost all samples, the particles analyzed are characterized by high degrees of circularity and roundness, while the aspect ratio values suggest a relatively low-to-moderate degree of elongation. The high values of roundness and the degree of elongation indicate that the inclusions are probably composed of quartz and calcite, the surface of which has been abraded by transportation processes (either by wind or water at the time of deposition).

**Table 4.** Mean and median values of several parameters from frontal and longitudinal tomographic sections of the selected samples.

| Sample | Section | Circularity | | Aspect Ratio | | Roundness | |
|--------|---------|------|--------|------|--------|------|--------|
| | | Mean | Median | Mean | Median | Mean | Median |
| P0006 | frontal | 0.96 | 1.00 | 1.47 | 1.46 | 0.73 | 0.68 |
| | longitudinal | 0.87 | 0.90 | 1.38 | 1.36 | 0.75 | 0.74 |
| P0023 | frontal | 0.85 | 0.88 | 1.57 | 1.47 | 0.69 | 0.68 |
| | longitudinal | 0.87 | 0.94 | 1.34 | 1.25 | 0.79 | 0.80 |
| P0046 | frontal | 0.94 | 0.99 | 1.39 | 1.29 | 0.76 | 0.77 |
| | longitudinal | 0.92 | 0.99 | 1.37 | 1.28 | 0.76 | 0.73 |
| P0073 | frontal | 0.77 | 0.81 | 1.75 | 1.60 | 0.62 | 0.63 |
| | longitudinal | 0.76 | 0.77 | 1.81 | 1.64 | 0.61 | 0.61 |
| P0149e | frontal | 0.71 | 0.76 | 1.65 | 1.67 | 0.63 | 0.60 |
| | longitudinal | 0.80 | 1.00 | 1.65 | 1.34 | 0.67 | 0.74 |
| P0173a | frontal | 0.63 | 0.64 | 2.12 | 1.89 | 0.54 | 0.53 |
| | longitudinal | 0.69 | 0.70 | 1.96 | 1.81 | 0.56 | 0.55 |

As shown in Table 3, a trend of decreasing circularity and roundness, slightly correlated with an increase in aspect ratio is observed. This suggests that the increase in frequency for particles with a more sub-angular shape may be well correlated with the previous tendencies observed in the case of the $\phi$ values.

### 3.2. Voids Analysis

Observing voids and inclusion morphology, size, distribution, and orientation can present valuable information on paste recipes and forming techniques [10,12]. The voids observed in the pottery sherds can have various natures, such as pores, joints, cracks, or burnt organics. They appear as low-density zones in the tomographic images, usually represented by black spots, and are thus easy to segment and process for digital analysis (Figures 4, 5, and S3–S6).

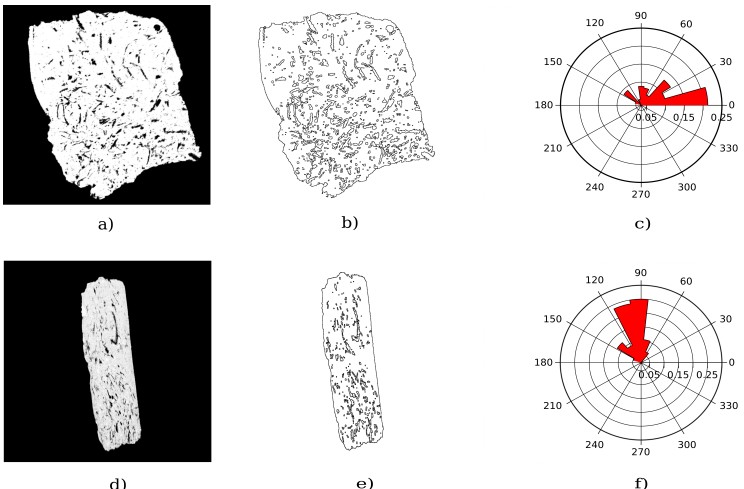

**Figure 4.** Up: segmented (**a**) and convolved (**b**) images of the P0083 sample; (**c**) rose diagram showing the preferred orientations of the voids. Down: segmented (**d**) and convolved (**e**) images of the analyzed side-view tomographic section; (**f**) rose diagram showing the preferred orientations.

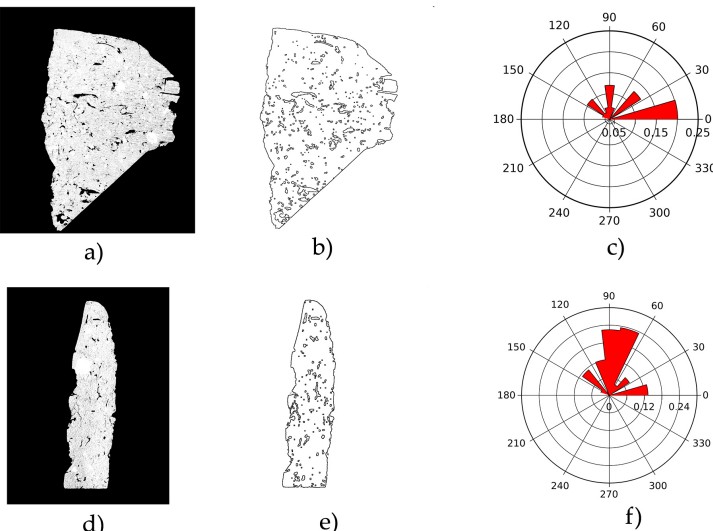

**Figure 5.** Up: segmented (**a**) and convolved (**b**) images of the P0173a sample; (**c**) rose diagram showing the preferred orientations of the voids. Down: segmented (**d**) and convolved (**e**) images of the analyzed side-view tomographic section; (**f**) rose diagram showing the preferred orientations.

Figures 4 and 5 show examples of the processed images used for the analysis and rose histograms with the corresponding preferred orientations of the voids. The plots and

subsequent analyses were performed for particles with an aspect ratio greater than 1 and a minimum Feret diameter greater than the reconstructed pixel size. It can be seen that all samples analyzed in this study show preferred orientations of voids due to the specific techniques applied during the manufacturing process.

Table 5 shows the mean and circular standard deviation values calculated for the images taken from frontal and longitudinal views of the tomographic reconstructions. In several cases, such as P0046 or P0147, the most probable orientations closely follow the mean value calculated. This feature is also well emphasized by the low value for the circular standard deviation determined for these potsherds.

The Rayleigh z test values calculated for each sample are between 29.63 and 264.05, while the critical ones are between 2.986 and 2.994 (for an $\alpha$ level of 0.05). The differences between the values determined for each sample and the critical ones emphasize the preferred orientations of the voids.

**Table 5.** Mean and circular standard deviation (CSD) values for the angles from the frontal and longitudinal tomographic sections.

| Sample | Section | Mean Angle (°) | Circular Standard Deviation (°) |
|--------|---------|---------------|--------------------------------|
| P0023 | frontal | 76.64 | 68.79 |
| | longitudinal | 84.15 | 32.77 |
| P0040 | frontal | 50.80 | 57.91 |
| | longitudinal | 87.77 | 37.52 |
| P0046 | frontal | 90.82 | 34.99 |
| | longitudinal | 84.22 | 36.04 |
| P0083 | frontal | 67.36 | 59.08 |
| | longitudinal | 258.16 | 32.26 |
| P0147 | frontal | 261.93 | 42.43 |
| | longitudinal | 262.41 | 34.35 |
| P0149e | frontal | 50.88 | 61.74 |
| | longitudinal | 63.04 | 60.98 |
| P0173a | frontal | 85.22 | 62.27 |
| | longitudinal | 75.66 | 41.07 |

## 4. Discussion

### 4.1. Clay Selection and Preparation

Regardless of the periods and sites, the sorting degree of the clay used for pottery was similar for all the analyzed samples, indicating that the strategies for choosing the clay sources as well as the processing methods of the raw materials remained similar for over two millennia; the small variations that can be observed may result from the particularities of the local geological background around the production sites (e.g., Spataro [23] for Starčevo-Criș pottery). The well-defined mineral inclusions in the 2D sections had similar roundness and sphericity, specific to natural inclusions such as sand, pebbles, and calcareous concretions. The only cases where the nature of such inclusions can be assumed is when collapsed calciferous particles were observed, which occurred frequently in the samples from the Chalcolithic period (e.g., P0106 and P0108 from Sultana-Malu Roşu site).

In the case of the pottery with incised/excised decoration from the Middle Neolithic and Early Chalcolithic, an examination of the longitudinal sections of the sherds pointed out that two different paste recipes were used: a poorly sorted, heavily tempered one (with organic, grog, or both of them) was used during the primary forming of the vessel's shape, whereas the external surface was prepared for decoration by applying a consistent layer (1–3 mm) of a fine, well sorted, clay mixture. This technological choice was most probably a practical one, as the complex patterns created through incision and excision are hard to perform on a clay with large impurities.

## 4.2. Pottery Tempering

The identification and analysis of tempers were not the main objectives of our study, as a differentiation via an automated segmentation procedure between natural mineral inclusions and intended added ones (e.g., tempers) is hard to perform on tomographic images, as the process involves a lot of user interaction and sophisticated filtering processes, which limits its practicability [10]. However, a close inspection of the generated images made it clear that organic tempering during the Early Neolithic and grog tempering during the Early and Developed Chalcolithic constituted two distinct and consistent technological traditions. It appears that during the Middle Neolithic, the choice of tempers was more variable, as both organic and grog tempering were used, sometimes for making similar pots at the same site (e.g., P0082 and P0083) or mixed (together) in paste recipes (e.g., P0173a).

The presence of organic tempering was obvious in many of the analyzed images, as the voids created by the plants fragments that were lost during the firing process have specific shapes and dimensions. The voids left by vegetal temper were segmented and convolved by digital processing in void orientation analysis, thus emphasizing an abundance that ranges between 10% and 20%. Based on the similarities observed in the data and those from former studies on the subject [23,28], it can be supposed that cereal chaff was the source of organic tempering in the analyzed sherds.

## 4.3. Primary Forming Techniques

There exists a scholarly consensus regarding the absence of the potter's wheel during the Neolithic and Chalcolithic periods, implying that pots were formed manually by different techniques such as pinching, coiling, slabbing, and molding, or by combining more than one of these methods (e.g., [8]). Clues indicative of the use of these techniques may be found within the internal structure of the samples through visual inspection of inclusions (that appear with different shapes, sizes, and shades of gray) and/or voids (which can be seen as black areas), with seemingly random orientations in some cases and/or more clear ones with preferred directions in other cases [12]. By coupling, segmenting, and convolving images and creating rose diagrams from them, the preferred and/or dispersed orientation of the voids is made clearer and considerably easier to identify, which facilitates better overall imaging and improves the interpretation of the results.

Morphological analysis of the voids and the images acquired through the process has proven indispensable in the interpretation of the primary shaping techniques; in effect, during the forming and afterwards firing of a vessel, voids form within the internal structure, indicating the last actions that influenced their dimensions, shapes, and, most importantly, orientation. Hence, based on our results, primary forming techniques such as coiling and slab building were the most conspicuously used, with the vertical or horizontal orientations of the voids constituting strong evidence supporting this hypothesis. Meanwhile, according to mechanical shock logic, which occurs when the clay is beaten at a certain pace while forming a vessel, moulding can be observable when the voids and/or inclusions are dispersed [8], without the trace of any distribution patterns that would suggest preferred orientations. However, it should be considered that the use of secondary forming techniques (shaping and thinning of vessel's walls by beating, trimming, and other techniques) and the application of certain surface treatments (scraping, smoothing, burnishing, and polishing) [37] can also influence the distribution and orientation of voids and inclusions on the surface of the samples. Consequently, the core of the sherds is the most suitable for this type of analysis, as it retains clear traces of the initial shaping through primary forming techniques [38]. Based on these principles, the combination of multiple techniques for forming the same vessel could also be visible, as, within the same sample, both preferred (vertical and/or horizontal) and/or dispersed orientations of voids and inclusions were identified. Nevertheless, it is worth mentioning that if multiple forming techniques are identified within the core of the same sherd, an interpretation designating a single forming technique may be erroneous, as a sherd generally represents but a small percentage of the whole pot, and merely offers a partial view of the forming process.

## 5. Conclusions

The presented work focused on the non-destructive investigation of pottery fragments specific to the Neolithic and Chalcolithic periods in Southern Romania. XCT was used to analyze the samples and to reveal multiple elements from the internal structure. Moreover, digital image analysis was applied to characterize features such as voids and inclusions which were segmented by means of specific filtering techniques, such as segmentation and convolution.

Firstly, particles with higher grayscale values and those presenting specific shapes were selected for dimensional analysis with the phi function of the MinFeret diameter and several shape descriptors. The results revealed particles that belonged within the size range specific to fine sand. Moreover, in most cases, particles appeared as moderately and poorly sorted.

Secondly, voids were separated from the sample by applying a gamma filter followed by convolution for edge enhancement. Data provided as angles were plotted as rose diagrams to emphasize preferred orientations. They suggested the use of specific techniques for pottery forming, such as coiling and slab building. Limitations regarding the interpretation of primary forming techniques based on inclusions and voids orientation were highlighted, since a sherd represents only a modest percentage of the pot; as a consequence, it may only provide but a glimpse of the whole story of a potter's technological choices and practices. Furthermore, secondary forming techniques and surface finishing could also have an impact on the orientation of the partices, which is why such a research endeavor must focus on the core of a sample.

In spite of the advantages associated with the use of a non-destructive technique in the technical analysis of artifacts, this approach also has some drawbacks. For instance, finer particles and voids that may have given additional information about the scanned fragments were excluded from the analysis, since their dimensions were comparable to the ones of the tomographic reconstructions. However, due to the current pace in technological progress, we can conclude that XCT scanning has some excellent potential for revealing an increasing number of aspects of past material culture without damaging the samples being analyzed. In particular, XCT scanning of ceramic sherds, which are low-density materials, can offer clear 3D reconstructions of their internal structure.

Ultimately, this work represents another step forward in the use of this method in ceramic technological studies. This field of research, which is constantly developing and deals with increasingly larger datasets, contributes to broadening our understanding of what ceramics manufacturing may have meant within different well-defined spatial–temporal frameworks.

**Supplementary Materials:** The following supporting information can be downloaded at: https://www.mdpi.com/article/10.3390/heritage6100347/s1, Figures S1–S6.

**Author Contributions:** Conceptualization, D.S., O.G.D., V.O. and B.M.; methodology, D.S., V.V., P.M., V.P. and M.D.; validation, D.S., O.G.D. and B.M.; formal analysis, D.S.; writing—original draft preparation, D.S., V.O. and B.M.; writing—review and editing, B.M., C.L., V.O. and I.M.; project administration C.L.; funding acquisition, C.L. All authors have read and agreed to the published version of the manuscript.

**Funding:** The work of University of Bucharest team was supported by a grant from the Ministry of Research, Innovation, and Digitization, contract number 41PFE/30.12.2021, within PNCDI III. V.O. was supported by a Research Fellowship for Visiting Professors (6247/08.06.2023) within the Research Institute of the University of Bucharest (ICUB) under the project "The rise and fall of the grog-tempered pottery in the Southern Romania during the Chalcolithic period (5th millennium BCE)". C.L. was funded by a FDI-2023 grant from CNFIS, contract number 0609/2023.

**Data Availability Statement:** The data presented in the study are available on request from corresponding author.

**Acknowledgments:** We thank Theodor Ignat (Bucharest Municipality Museum, Romania) and Daniela Dimofte (University of Bucharest, Romania) for the technical, scientific, and logistical support granted in 2017–2018 for collecting and selecting the sherds batch used in this study.

**Conflicts of Interest:** The authors declare no conflict of interest.

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
