# Peer review of "Non-Destructive Textural Characterization of Southern Romanian Neolithic and Chalcolithic Pottery Using Digital Image Analysis on Tomographically Reconstructed Sections"

_heritage, doi:10.3390/heritage6100347_

Round 1

Reviewer 1 Report

In the section 2.2 improve and clarify the description of the instrument and the measurement condition used, for example the voxel size..

In the section 3.1 is not clear the archaeometrical information taken from the statistical parameters used, please revise accurately this aspect and if the information it is not clear please said or remove the paragraph.

The table 3 is not clear, please revise it and re-write it. 

Finally, in my opinion is preferable avoid sensationalism in the titles, moreover the title is not representative of the work, please consider a review of it.

Author Response

Dear Reviewer, 

Thank you for your comments. Most of the highlights have been made using blue color, in several exceptions a red highlight was performed, since similar information was requested by other reviewers.  Below please find the answers as well as the manuscript attached with the highlights performed accordingly: 

In the section 2.2 improve and clarify the description of the instrument and the measurement condition used, for example the voxel size.

Answer:

The description of the instrument and the ones regarding the measurement conditions were clarified. Table 2 highlights the voxel size for each sample, while the conditions regarding the voltage and current used are highlighted in red at the lines 123 – 124: ‘The scans on potsherds were performed using the following parameters: 100-kV and 300-μA beam, 354-ms exposure time.’ 

In the section 3.1 is not clear the archaeometrical information taken from the statistical parameters used, please revise accurately this aspect and if the information it is not clear please said or remove the paragraph.

Answer:

The archaeometrical information in section 3.1 was revised by introducing additional phrases in blue color, in order to make a connection between the results and the discussion, between lines 218 and 222: “Moreover, several trends are observed. In the analyzed samples a decrease in the mean Phi(MinFeret) is generally observed, while sorting degree presents an increasing trend, especially in the case of the frontal tomographic sections. This may suggest the existence of changes in the type of clay source used, with a slight tendency towards the use of clays having coarser and more poorly sorted distribution of inclusions.”

The other one is inserted between lines 230 and 233: “Like in the case of table 3, a trend of decreasing circularity and roundness, slightly correlated with an increase of aspect ratio is observed. This suggests the increase in frequency for particles having a more sub-angular shape, this may be well correlated with the previous tendencies observed in the case of the φ values.”

Also, the archaeometric information may be limited since we did not make any use of mineralogical and chemical investigations.

The table 3 is not clear, please revise it and re-write it.

Answer:

Since an additional table, table 3 is now labeled as table 4. It was revised by emphasizing better the headers for each shape descriptor in blue color above each column representing the mean value.

Finally, in my opinion is preferable avoid sensationalism in the titles, moreover the title is not representative of the work, please consider a review of it.

Answer:

The title of the work has been revised by removing the phrase ‘X-ray ability sees through the past!’

Reviewer 2 Report

General comments

23 samples to cover the whole spectrum: how is this made sure; based on what criteria is it clear that this is representative to cover research question over the entire population of ceramic artefacts?  

The voxel size of the images in unclear, as well as size distribution of the smallest range of analysed particle inclusions. There must be at least a factor 3 difference in the size to make sensible results. These values are however not clear from the text. 

The main discussion is that the sorting degree of the clay used for pottery was similar over two centuries (regarding clay sources and processing methods). However, it can be questioned if particle analysis of dense inclusions and analysis of macropores/voids is sufficient to draw this conclusion. peer-reviewed study indicating that particle analysis of dense inclusions is a proven determinant for the clay resource would be necessary to support this; and to my knowledge this is not existing. Chemical and mineralogical analysis would make a stronger support for this case.

Also in production technology (section 4.3), papers using both X-rays as well as other microscopic methods, show that the texture of the matrix and orientation different types of inclusions, and not solely voids, are indicative for processing methods.

Specific comments

-          The main technique, X-ray Computed Tomography, is a conceptual technique and does not need to have an article in front of it. Use ‘X-ray Computed Tomography’ instead of ‘the X-Ray Computed Tomography’.

-          An abbreviation for X-ray Computed Tomography is common; consider the use of an abbreviation throughout the text instead of a combination of full words like ‘X-ray Computed Tomography’ or ‘computed tomography’.

-          Some references are poor and/or very old, where more accurate and state-of-the-art references are in place, e.g. for describing recent advances in X-ray CT (ref 3,4), ref. 15 would be perfectly suited, as well as other more recent publications.

-          Line 24: chaîne opératoire is sufficiently embedded as a term in the scientific community, so that it does not need italic highlighting as foreign language.

-          Line 56-60: it is preferred to use ‘voxel resolution’ or ‘voxel size’ as terminology in this section, to make clear this is about the size of de voxels, which is not the same as general ‘resolution’. Also line 58 is untrue; also the dimension of particles with the size of multiple voxels can be affected by limited voxel size, due to the partial volume effect. Use the correct terminology.

-          Southern: if not a formal geographic name (Southern Romania), than write without capital letter.

-          Fig. 1 needs a more specific location referencing (e.g. on a larger scale map). Additionally, the size of orientation and scale bar should be checked for readability in the final format.

-          Line 82: ‘no preferences for certain periods’ is ambiguous. Rephrasing is advisable.

-          Line 91: ‘imagistic’ is not a correct term; imaging is.

-          Line 97-102 is methods rather than objectives; consider shifting this section.

-          Line 102: replace Ref. with the author names; also the reference in the list is wrong.

-          Line 107: Table number is not given.

-          Section 2.2: what are the specific parameters for data acquisition? The system specifics are given, but not the experimental settings. Additionally, what is the resulting 3D voxel size of the images?

-          Line 136: what are representative samples?

-          Line 136: what approach was taken to ‘threshold’ the image? This is a segmentation process, it is advisable to use this terminology.

-          Line 156: ‘projection’ is a word typically used for radiographs; ‘visualizing of virtual projections’ = ‘rendering [gave insights…’]

-          Line 157: consider using gray value instead of shade, or better, refer to the underlying physics and consider using attenuation.

-          Linke 158: ‘most fabrics could be interpreted as particles’; a fabric is not a particle.

-          Line 168/methods: is the dimensional particle analysis performed in 2D or 3D? This is unclear.

-          Line 172 + 197 + 216: Table ?? is not defined.

-          Line 206: building techniques is maybe not the correct word. Consider using production technology or historical techniques.

-          Line 207: what are clay pores? If these are the pores in the clay matrix, then it is very unlikely that these can be observed -let alone isolated, line 209- in traditional X-ray imaging.

-          Line 209: consider using segmentation as instead of for ‘isolate’.

In general a language check is advisable, mainly focusing on terminology and the use of single words like ‘tricky’, ‘great worth’ (value), which are sometimes inappropriate.

Author Response

Dear reviewer,

Thank you for your comments and suggestions. Below please find the answers to each comment and the modified manuscript with the highlights, which were performed using red color: 

23 samples to cover the whole spectrum: how is this made sure; based on what criteria is it clear that this is representative to cover research question over the entire population of ceramic artefacts?

Answer:

In this case we used ceramic traditions as a synonym for archaeological cultures, as the Neolithic and Chalcolithic cultures from Southern Romania were mainly defined based on the decorative styles present of pottery. Form this point of view, our selection covers the “whole spectrum”, all the archaeological cultures from the study area being represented by a least one sherd in the selected batch. To make the things more clearly for the reader “ceramic traditions” will be replaced with “archaeological cultures”.

The voxel size of the images in unclear, as well as size distribution of the smallest range of analysed particle inclusions. There must be at least a factor 3 difference in the size to make sensible results. These values are however not clear from the text.

Answer:

Modifications were performed regarding it. An additional table (Table 2) was made, in which the voxel size is highlighted for each sample. Modifications of the data were performed and all the particles having a size of at least 3 times greater than the voxel size were considered, while the rest of them were rejected. This is also highlighted in the section 2.3, between the lines 155 and 157: “Only inclusions having a MinFeret diameter at least 3 times greater than the voxel size were selected, to ensure the final data are not affected by the XCT and segmentation-generated artifacts”. The new data are highlighted with red color in tables 3 and 4. Therefore, there is no need to emphasize the distribution of smallest particles, since their dimension is more or less comparable to the one of the voxel size, and they may alos be generated as artifacts due to the segmentation process.

Consequently, phrases which described the old data were removed, as the one from lines 189 – 191, while between lines 209 and 217 new information was introduced to interpret the data: ‘as the values obtained through the method of moments have a greater variability within all samples than the ones obtained through the graphic method. The values obtained through the method of moments emphasize the existence of extremely leptokurtic distributions in the case of P0023, P0073 and P0173a, while the rest are framed as platykurtic and mesokurtic. Graphic method values shows that all samples present platykurtic to mesokurtic distributions of particles.’ A similar modification was performed in the case of the degree of elongation of the aspect ratio at line 226, to have a better correspondence to the new data.

The main discussion is that the sorting degree of the clay used for pottery was similar over two centuries (regarding clay sources and processing methods). However, it can be questioned if particle analysis of dense inclusions and analysis of macropores/voids is sufficient to draw this conclusion. peer-reviewed study indicating that particle analysis of dense inclusions is a proven determinant for the clay resource would be necessary to support this; and to my knowledge this is not existing. Chemical and mineralogical analysis would make a stronger support for this case.

Answer:

The sorting degree of the clays used for pottery was estimated considering the density and distribution of both voids and non-plastics observed by the close inspection of the XTC images. Also, our conclusion was backed by the dimensional and sorting degree analysis (Table 3) of the outlined inclusions by Fiji. However, we agree that at least one complementary chemical or mineralogical analysis would have made this assertion more accurate.

Also in production technology (section 4.3), papers using both X-rays as well as other microscopic methods, show that the texture of the matrix and orientation different types of inclusions, and not solely voids, are indicative for processing methods.

Answer:

Totally agree, and that is why the role of the non-plastics orientations in determining pottery forming techniques was emphasized in the first paragraph of the section 4.3 (see also the first paragraph of section 3.2). However, the voids preferred orientations were more visible after the inspection of the XCT images, especially in the sherds tempered with vegetal remains, making them more suitable for being integrated in the rose diagram analysis

Specific comments

  1. - The main technique, X-ray Computed Tomography, is a conceptual technique and does not need to have an article in front of it. Use ‘X-ray Computed Tomography’ instead of ‘the X-Ray Computed Tomography’.

    - Answer: The articles in the front of the expression ‘X-ray Computed Tomography’, by highlighting it through red color and a strikethrough.

  2. - An abbreviation for X-ray Computed Tomography is common; consider the use of an abbreviation throughout the text instead of a combination of full words like ‘X-ray Computed Tomography’ or ‘computed tomography’.

    - Answer: ‘XCT’ is now used as an abbreviation and highlighted instead of ‘X-ray Computed Tomography’ or ‘X-ray CT’.

  3. - Some references are poor and/or very old, where more accurate and state-of-the-art references are in place, e.g. for describing recent advances in X-ray CT (ref 3,4), ref. 15 would be perfectly suited, as well as other more recent publications.

    - Answer: Some additional references regarding some advances of X-ray CT were introduced between lines 394 and 397:

5. Beister, M., Kolditz, D & Kalender, W. A. Iterative reconstruction methods in X-ray CT. Physica Medica, 2012 28(2), 94-108, https://doi.org/10.1016/j.ejmp.2012.01.003

6. Hsieh, J., Nett, B., Yu, Z., et al. Recent advances in CT image reconstruction, Current Radiology Reports, 2013, 1, pp.39-51, https://doi.org/10.1007/s40134-012-0003-7

  1. - Line 24: chaîne opératoire is sufficiently embedded as a term in the scientific community, so that it does not need italic highlighting as foreign language.

    - Answer: We have performed the modification and highlighted the term in red accordingly, without the use of italic.

  2. - Line 56-60: it is preferred to use ‘voxel resolution’ or ‘voxel size’ as terminology in this section, to make clear this is about the size of de voxels, which is not the same as general ‘resolution’. Also line 58 is untrue; also the dimension of particles with the size of multiple voxels can be affected by limited voxel size, due to the partial volume effect. Use the correct terminology.

    - Answer: The term ‘voxel size’ is used and highlighted at line 61. At line 63, we have modified and highlighted that also the particles with dimensions comparable to the voxel size are hard to be dimensionally characterized.

  3. - Southern: if not a formal geographic name (Southern Romania), than write without capital letter.

    - Answer: Since it is a formal geographic name, it is written with capital letter.

  4. - Fig. 1 needs a more specific location referencing (e.g. on a larger scale map). Additionally, the size of orientation and scale bar should be checked for readability in the final format.

    - Answer: Fig. 1 has a better location referencing and the size of orientation and sale bar were checked for readability, being readable in the final format.

  5. - Line 82: ‘no preferences for certain periods’ is ambiguous. Rephrasing is advisable.

    - Answer: Regarding the manufacturing methods, there are no comprehensive studies for observing specific choices for every archaeological culture and the tendencies over time.

  6. - Line 91: ‘imagistic’ is not a correct term; imaging is.

    - Answer: The modification was performed, ‘imaging’ is highlighted with red color.

  7. - Line 97-102 is methods rather than objectives; consider shifting this section.

    - Answer: The phrases between the lines 97 and 102 was shifted at section 2.3: ‘In the case of the graphic method, the formulas used to calculate the statistical parameters (standard deviation, skewness, and kurtosis) are the ones developed by Folk and Ward [32]. In contrast, the MinFeret diameter provided by the ImageJ analysis was used for phi scale, as suggested by Dal Sasso et al.[19].’

  8. - Line 102: replace Ref. with the author names; also the reference in the list is wrong.

    - Answer: The expression ‘ref.’ was replaced with Dal Sasso et al.

  9. - Line 107: Table number is not given.

    - Answer: The modification was performed and the Table 1 is defined and highlighted (line 109).

  10. - Section 2.2: what are the specific parameters for data acquisition? The system specifics are given, but not the experimental settings. Additionally, what is the resulting 3D voxel size of the images?

    - Answer: The specific parameters were added at the lines 123 – 124: ‘The scans on potsherds were performed using the following parameters: 100-kV and 300-μA beam, 354-ms exposure time.’ Additionally, Table 2 provides the voxel size for each scanned sample and is highlighted accordingly.

  11. - Line 136: what are representative samples?

    - Answer: The representative samples were selected on the basis of the large number of particles left after the segmentation process. A phrase was inserted at line 142 – 143.

  12. - Line 136: what approach was taken to ‘threshold’ the image? This is a segmentation process, it is advisable to use this terminology.

    - Answer: The modification was performed and ‘segmentation’ is highlighted accordingly.

  13. - Line 156: ‘projection’ is a word typically used for radiographs; ‘visualizing of virtual projections’ = ‘rendering [gave insights…’]

    - Answer: The word ‘section’ is used and highlighted accordingly instead of ‘projection’.

  14. - Line 157: consider using gray value instead of shade, or better, refer to the underlying physics and consider using attenuation.

    - Answer: The expression ‘gray shade’ was modified with ‘grayscale value’, being highlighted.

  15. - Linke 158: ‘most fabrics could be interpreted as particles’; a fabric is not a particle.

    - Answer: The word ‘fabrics’ was replaced with ‘non-plastics’.

  16. - Line 168/methods: is the dimensional particle analysis performed in 2D or 3D? This is unclear.

    - Answer The dimensional particle analysis was performed in 2D, since the tomographic sections visualized were exported as TIFF files, this being emphasized by the expression ‘2D’ found at line 129 and ‘2D sections’ at line 180.

  17. - Line 172 + 197 + 216: Table ?? is not defined.

    - Modifications were performed at the tables from the lines 125, 187, 223 and 247 and they were highlighted.

  18. - Line 206: building techniques is maybe not the correct word. Consider using production technology or historical techniques.

    - ‘Building technique’ was replaced with ‘forming techniques’ and highlighted at line 236.

  19. - Line 207: what are clay pores? If these are the pores in the clay matrix, then it is very unlikely that these can be observed -let alone isolated, line 209- in traditional X-ray imaging.

    - The word ‘clay’ was removed.

  20. - Line 209: consider using segmentation as instead of for ‘isolate’.

    - The word ‘segmentation’ is used instead of ‘isolate’ and it is highlighted at line 239.

Comments on the Quality of English Language

In general a language check is advisable, mainly focusing on terminology and the use of single words like ‘tricky’, ‘great worth’ (value), which are sometimes inappropriate.

Answer:

A language check was performed on the terminology and words such as ‘tricky’ and ‘great worth’ were removed.

Reviewer 3 Report

This article presents interesting results, but more data would be useful to understand them. More information about the experimental setup, such as the voltage used for the data acquisition and distance between sample and detector, should be added, as well as the final spatial resolution obtained with setup used.

The considerations about the limitation of the CT approach (46-60) are correct for a conventional setup but not for a synchrotron radiation approach, where the monochromaticity and the shape of the beam allow to reduce the presence of artifacts, while the spatial coherence of the source contribute to a better definizion of the different phases inside the sample.

A minor editing of English language is required.

A minor editing of English language is required.

Author Response

Dear reviewer, 

Thank you for your comments and suggestions. Please find below the answers to your request. They have been highlighted in the manuscript using green color, while in some cases, red color was used, since there were similar requests asked by other reviewers: 

This article presents interesting results, but more data would be useful to understand them. More information about the experimental setup, such as the voltage used for the data acquisition and distance between sample and detector, should be added, as well as the final spatial resolution obtained with setup used.

Answer:

The information about the experimental setup was improved by adding the applied voltage and current at the lines 123 – 124: ‘The scans on potsherds were performed using the following parameters: 100-kV and 300-μA beam, 354-ms exposure time.’ Additionally, Table 2 emphasizes the reconstructed voxel size and the source-to-object distance. The phrase and the Table 2 are highlighted with red color, while column 3 of the Table 2 is highlighted in green.

The considerations about the limitation of the CT approach (46-60) are correct for a conventional setup but not for a synchrotron radiation approach, where the monochromaticity and the shape of the beam allow to reduce the presence of artifacts, while the spatial coherence of the source contribute to a better definizion of the different phases inside the sample.

Answer:

Since the current research was performed using a conventional CT, the word ‘conventional’ was highlighted for a more clear delimitation from the CT performed using synchrotron radiation.

A minor editing of English language is required.

Answer:

Editing of English language was performed.

Round 2

Reviewer 2 Report

The authors have taken the comments into consideration and have made changes accordingly.

I would still suggest to contextualize the outcomes a bit more. For example, if chemical or mineralogical analysis could be of added value, then why not state this, either as a future outlook, either as situating XCT as a complementary technique. 

The language is overall ok, but conciseness and fluency of reading can be improved.